Comprehensive analysis of lncRNA-associated competing endogenous RNA network in tongue squamous cell carcinoma

Zhang Shusen 1 2
Cao Ruoyan 1
Li Qiulan 3
Yao Mianfeng 4
Chen Yu 1
Zhou Hongbo zhb2540@csu.edu.cn 1
1 Department of Prosthodontics, Xiangya Stomatological Hospital & School of Stomatology, Central South University , Changsha , China
2 Department of Stomatology, Hunan University of Medicine , Hunan , China
3 Department of Stomatology, The Second Xiangya Hospital, Central South University , Changsha , China
4 Department of Stomatology, Xiangya Hospital, Central South University , Changsha , China
Khan Imran
Electronic publication date: 2019 Feb 6
Publication date: 2019
Volume: 7
Electronic Location ID: e6397
Received 2018 Oct 3; Accepted 2019 Jan 7
Copyright: ©2019 Zhang et al.
Copyright year: 2019
Copyright holder: Zhang et al.
License: This is an open access article distributed under the terms of the Creative Commons Attribution License, which permits unrestricted use, distribution, reproduction and adaptation in any medium and for any purpose provided that it is properly attributed. For attribution, the original author(s), title, publication source (PeerJ) and either DOI or URL of the article must be cited.
License URL: https://creativecommons.org/licenses/by/4.0/

Keywords: Tongue squamous cell carcinoma, Long noncoding RNA, Prognosis, Competing endogenous RNA network

Funding: The authors received no funding for this work.

==============================
Background

Increasing evidence has demonstrated that long non-coding RNAs (lncRNAs) play an important role in the competitive endogenous RNA (ceRNA) networks in that they regulate protein-coding gene expression by sponging microRNAs (miRNAs). However, the understanding of the ceRNA network in tongue squamous cell carcinoma (TSCC) remains limited.

Methods

Expression profile data regarding mRNAs, miRNAs and lncRNAs as well as clinical information on 122 TSCC tissues and 15 normal controls from The Cancer Genome Atlas (TCGA) database were collected. We used the edgR package to identify differentially expressed mRNAs (DEmRNAs), lncRNAs (DElncRNAs) and miRNAs (DEmiRNAs) between TSCC samples and normal samples. In order to explore the functions of DEmRNAs, Kyoto Encyclopedia of Genes and Genomes (KEGG) pathway analysis was performed. Subsequently, a ceRNA network was established based on the identified DElncRNAs–DEmiRNAs and DEmiRNAs–DEmRNAs interactions. The RNAs within the ceRNA network were analyzed for their correlation with overall disease survival. Finally, lncRNAs were specifically analyzed for their correlation with clinical features in the included TSCC patient samples.

Results

A total of 1867 mRNAs, 828 lncRNAs and 81 miRNAs were identified as differentially expressed in TSCC tissues (—log 2fold change— ≥ 2; adjusted P value <0.01). The resulting ceRNA network included 16 mRNAs, 56 lncRNAs and 6 miRNAs. Ten out of the 56 lncRNAs were found to be associated with the overall survival in TSCC patients (P < 0.05); 10 lncRNAs were correlated with TSCC progression (P < 0.05).

Conclusion

Our study deepens the understanding of ceRNA network regulatory mechanisms in TSCC. Furthermore, we identified ten lncRNAs (PART1, LINC00261, AL163952.1, C2orf48, FAM87A, LINC00052, LINC00472, STEAP3-AS1, TSPEAR-AS1 and ERVH48-1) as novel, potential prognostic biomarkers and therapeutic targets for TSCC.

Introduction

Tongue squamous cell carcinoma (TSCC) is the most common type of oral squamous cell carcinoma (OSCC) with remarkable invasiveness, early lymph node metastasis and a poor prognosis (Sano & Myers, 2007; Zhou et al., 2015). The quality of life for TSCC survivors is often reduced due to speech disfunction, mastication, and deglutition. Recently, the incidence and mortality of TSCC have steadily risen in the United States (Siegel et al., 2014; Siegel, Miller & Jemal, 2015; Siegel, Miller & Jemal, 2016; Siegel, Miller & Jemal, 2017; Siegel, Miller & Jemal, 2018). Despite significant advancements in surgical excision, radiotherapy and chemotherapy, mortality rates and recurrence rates for this cancer remain high (Adeel & Suhail, 2016; Chen et al., 2018). For these reasons, the molecular mechanisms of TSCC tumorigenesis urgently require further study; potential biomarkers as well as therapeutic targets in this cancer should be identified in order to improve clinical outcomes.

Long noncoding RNAs (lncRNAs) are a subclass of noncoding RNAs longer than 200 nucleotides (Ponting, Oliver & Reik, 2009). Of late, lncRNAs have been a new focus of cancer research and were found to be involved in tumorigenesis and metastasis (Chen et al., 2017; Gutschner et al., 2013; Yang et al., 2017; Yang et al., 2016b; Yuan et al., 2014). In TSCC, lncRNAs were reported to act as oncogenes or tumor suppressors and affect patient prognosis. For example, knockdown of AFAP1-AS1 could suppress cell proliferation, migration and invasion in TSCC (Wang et al., 2018). Additionally, the overexpression of lncRNA MEG3 inhibited cell proliferation and induced apoptosis in TSCC (Jia et al., 2014). The current literature has demonstrated that lncRNAs regulate gene expression via genetic imprinting, splicing regulation, chromatin remodeling, mRNA decay, and translational regulation (Zhu et al., 2013). However, the formation and development of tumors is a complex pathophysiological process. The mechanisms by which lncRNAs affect TSCC biology remain unelucidated.

Salmena et al. (2011) proposed the competing endogenous RNA (ceRNA) hypothesis, which stated that lncRNA could crosstalk with mRNA by sharing common microRNA response elements (MREs) with miRNA. More and more studies have validated the involvement of ceRNA crosstalk in the development and progression of various tumors, such as those of breast cancer, hepatocellular cancer and pancreatic cancer. Some of the few relevant studies on such crosstalk in TSCC have been verified, such as LINC00511/ miR-765/ LAMC2 (Ding, Yang & Yang, 2018) and H19/ let-7a/ HMGA2 (Kou et al., 2018). Furthermore, comprehensive analysis of TSCC-associated lncRNAs and miRNAs in a whole genome wide context is lacking, especially based on high-throughput sequencing with a large-scale sample size.

To better understand how lncRNAs regulate gene expression by sponging miRNAs in TSCC, we build a ceRNA network based on the TCGA database, including 16 mRNAs, 56 lncRNAs and six miRNAs. In addition, we found 10 lncRNAs to be associated with survival and 10 lncRNAs having an association with carcinogenesis. Results of these analyses are a starting point to analyze ceRNA crosstalk and gain insight into the molecular mechanisms participating in the tumorigenesis and progression of TSCC.

Materials and Methods

Patients and samples

RNA sequencing data and the corresponding clinical information for our TSCC dataset were retrieved from the TCGA data portal. The inclusion criteria were set as follows: (1) patients with follow-up survival times less than 2,000 days; (2) patients with detailed clinicopathological information including age, gender, survival time, survival status, pathological stage, TNM stage. As most patients were missing data about their metastatic states, we did not analyze this information. After filtering available data with our inclusion criteria, a total of 122 TSCC patients and 15 normal controls were included in our analysis. The clinical and pathological characteristics of the TSCC patients are summarized in Table 1. This study conformed with the publication guidelines provided by TCGA (https://cancergenome.nih.gov/publications/publicationguidelines) and as our data was obtained from TCGA database, approval by an ethics committee was not required.

Table 1 Clinicopathological characteristics of 122 patients with tongue squamous cell carcinoma

Characteristic	Subtype	No. of cases (%)	
Age (years)	<60	59 (48.4)	
	≥60	63 (51.6)	
Gender	Male	85 (69.7%)	
	Female	37 (30.3%)	
Pathologic stage	Stage I	13 (10.7%)	
	Stage II	19 (15.6%)	
	Stage III	30 (24.6%)	
	Stage IV	60 (49.1%)	
Pathologic T	T1	19 (15.6%)	
	T2	42 (34.4%)	
	T3	40 (32.8%)	
	T4	21 (17.2%)	
Pathologic N	N0	49 (40.2%)	
	N1	17 (13.9%)	
	N2	51 (41.8%)	
	N3	1 (0.8%)	
	NX	4 (3.3%)	
Vital status	Alive	72 (59%)	
	Dead	50 (41%)	

RNA sequence data processing

Level 3 RNASeq and miRNASeq data from TSCC samples up to June 30, 2018, including 122 TSCC tissues and 15 normal controls, were downloaded from the TCGA data portal. The sequence data originated from IlluminaHiSeq_RNASeq and IlluminaHiSeq_miRNASeq sequencing platforms; all the data are publicly available.

Analysis of differential expression profiles

The Ensembl database (http://www.ensembl.org/index.html, version 89) (Aken et al., 2016) was used to identify lncRNAs from the raw expression data. We discarded previously identified lncRNAs that were not included in this database. Differential expression analysis of mRNAs (DEmRNAs), miRNAs (DEmiRNAs) and lncRNAs (DElncRNAs) between TSCC and normal tissues was carried out using the edgeR package (Robinson, McCarthy & Smyth, 2010). For all p values, false discovery rate (FDR) was applied for multiple testing correction. Absolute log2FC ≥ 2 and the FDR <0.01 were used as cut-off criteria.

Functional enrichment analysis

In order to better understand the mechanisms involved in the tumorigenesis of TSCC, we conducted Gene Ontology (GO) functional enrichment analysis using DAVID (the Database for Annotation, Visualization and Integrated Discovery) with FDR <0.01 as the cut-off value. KEGG analysis was performed using the ClusterProfiler package in the R language with a cut-off value of adjusted p value <0.05.

Construction of ceRNA network

We used the miRcode database (Jeggari, Marks & Larsson, 2012) to predict lncRNA-miRNA interactions, which were then combined with selected miRNAs. Secondly, TargetScan (Fromm et al., 2015), miRTarBase (Chou et al., 2016) and miRDB (Wong & Wang, 2015) were used to retrieve and predict the targeted mRNAs of miRNA. In order to enhance the validity of this ceRNA network, we only included miRNA-targeted mRNAs present in all three databases and DEmRNAs. Finally, the ceRNA network was visualized using Cytoscape 3.6.1 software. A flowchart of the ceRNA network is presented in Fig. 1. We also performed smooth curve fitting to explore the relationship between ceRNA expression levels.

Figure 1 Flow chart of the ceRNA network construction.

Statistical analysis

For overall survival analysis, the log-rank test was employed to compare the difference between TSCC samples with different expresssion using Kaplan Meier survival curve. The cut-off point of expression was identified using survminer package (Li et al., 2018b). The edgeR package was used to screen out DElncRNAs associated with clinical features, by setting cut-off criteria of absolute log2FC ≥ 1 and the FDR <0.05. Unless specifically stated, a p value <0.05 was considered to represent statistical significance. All statistical analyses were performed using R software (version: 3.3.2).

Results

DEmRNAs in TSCC

According to the cut-off threshold of log2FC ≥ 2 and FDR <0.01, 717 (38.40%) up-regulated and 1,150 (61.60%) down-regulated genes were identified in TSCC (Table S1). Figure 2 shows the distribution of DEmRNAs between TSCC and normal controls. The expression heat map of DEmRNAs is shown in Fig. S1 . Red or green represents significantly upregulated and downregulated genes, respectively.

Figure 2 Volcano map of DEmRNAs.

Red spots represent up-regulated genes, and green spots represent down-regulated genes.

A total of 45 significantly enriched GO terms are listed in Table S2 that correspond to DEmRNAs. For “biological processes (BP)”, the top five terms were muscle filament sliding, collagen catabolic process, extracellular matrix organization, muscle contraction and skeletal system development; for the “cellular component (CC)” ontology the top five were, extracellular region, extracellular space, proteinaceous extracellular matrix, Z disc and collagen trimer; finally, the top five “molecular function (MF)” terms were, structural constituent of muscle, extracellular matrix structural constituent, calcium ion binding, heparin binding and cytokine activity (Fig. 3).

Figure 3 GO and KEGG pathway analyses.

(A) Top five cellular component terms of dysregulated genes in the GO analysis. (B) Top five biological processes terms of dysregulated genes in the GO analysis. (C) Top five molecular function terms of dysregulated genes in the GO analysis. (D) Top10 pathways of dysregulated genes in the pathway analysis. (E) Cytokine-cytokine receptor interaction map from KEGG analysis.

Additionally, a total of 20 significantly enriched KEGG pathways for the identified DEmRNAs are listed in Table 2, and the top 10 KEGG pathways are shown in Fig. 3D. The cytokine-cytokine receptor interaction pathway was found to harbor the largest number of DEmRNAs (Fig. 3E).

Table 2 Significantly enriched KEGG pathways regulated by DEmRNAs in tongue squamous cell carcinoma.

ID	Description	pvalue	p adjust	Count	
hsa04970	Salivary secretion	2.93E–12	8.40E–10	31	
hsa04974	Protein digestion and absorption	9.63E–11	1.38E–08	29	
hsa04512	ECM-receptor interaction	3.42E–08	3.27E–06	24	
hsa04060	CytokinE-cytokine receptor interaction	2.38E–06	0.000171	49	
hsa04020	Calcium signaling pathway	8.26E–06	0.000402	34	
hsa04510	Focal adhesion	8.41E–06	0.000402	36	
hsa00500	Starch and sucrose metabolism	2.25E–05	0.000924	12	
hsa04261	Adrenergic signaling in cardiomyocytes	5.94E–05	0.00213	27	
hsa05414	Dilated cardiomyopathy (DCM)	0.000147	0.004373	19	
hsa05410	Hypertrophic cardiomyopathy (HCM)	0.000152	0.004373	18	
hsa04976	Bile secretion	0.00022	0.00573	16	
hsa03320	PPAR signaling pathway	0.000363	0.008688	16	
hsa04973	Carbohydrate digestion and absorption	0.000833	0.018399	11	
hsa04971	Gastric acid secretion	0.001295	0.026541	15	
hsa04260	Cardiac muscle contraction	0.001957	0.035163	15	
hsa00910	Nitrogen metabolism	0.00196	0.035163	6	
hsa04964	Proximal tubule bicarbonate reclamation	0.002199	0.035586	7	
hsa04610	Complement and coagulation cascades	0.002232	0.035586	15	
hsa05146	Amoebiasis	0.002566	0.038758	17	
hsa00830	Retinol metabolism	0.003408	0.048901	13	

DElncRNAs in TSCC

Based on the cut-off criteria (log2FC ≥ 2 and FDR <  0.01), we identified 828 lncRNAs aberrantly expressed in TSCC compared to normal tissues, including 517 up-regulated (62.44%) and 311 down-regulated lncRNAs (37.56%) (Table S3). The distribution of all the DElncRNAs are presented as a volcano plot in Fig. 4 and an expression heat map of DElncRNAs is shown in Fig. S2.

Figure 4 Volcano map of DElncRNAs.

Red spots represent up-regulated genes, and green spots represent down-regulated genes.

DEmiRNAs in TSCC

To build our lncRNA-miRNA-mRNA ceRNA network, we also compared miRNA expression profiles in tumor tissues with normal tissues. In total, 81 DEmiRNAs were identified, including 42 up- and 39 down-regulated miRNAs (Table S4). A volcano plot of the related DEmiRNAs is shown in Fig. 5; a corresponding expression heat map is shown in Fig. S3.

Figure 5 Volcano map of DEmiRNAs.

Red spots represent up-regulated genes, and green spots represent down-regulated genes.

ceRNA network in TSCC

A dysregulated ceRNA network of lncRNA-miRNA-mRNA in TSCC was established based on the above data in order to better elucidate the role of DElncRNAs. First, the 828 DElncRNAs were retrieved from the miRcode, and 102 pairs of interacting lncRNAs and miRNAs were identified using the Perl language. Subsequently, we predicted that six DEmiRNAs could interact with 56 DElncRNAs. Then we found that these six DEmiRNAs targeted 221 mRNAs in all three databases (TargetScan, miRTarBase and miRDB). Among the 221 targeted mRNAs, only 16 mRNAs were found in the 1,867 DEmRNAs (Fig. S4). Finally, we constructed a ceRNA network relating to TSCC by incorporating 56 DElncRNAs, six DEmiRNAs and 16 DEmRNAs, as shown in Fig. 6. To confirm these findings, we performed smooth curve fitting between the expression levels of the DElncRNAs and DEmRNAs included in the ceRNA network. Our results indicated a positive correlation between ceRNA expression levels. For example, LINC00472 interacted with GREM2 mediated by mir-503 and SFTAIP regulated IL11 levels by sponging mir-211 (Fig. 7). We also contract GO and KEGG analysis to reveal the functions of the 16 DEmRNAs that were involved in the ceRNA network. Only two GO terms were significantly enriched (P < 0.05) (Table 3).

Figure 6 DElncRNAs mediated ceRNA regulatory network in TSCC.

The red nodes indicate expression up-regulation, and blue nodes indicate expression down-regulation. LncRNAs, miRNAs and mRNAs are represented by diamond, rounded rectangle, and ellipse, respectively.

Figure 7 Smooth curve fitting analysis.

(A) A smooth curve fitting for the relationship between LINC00472 and GREM2. (B) A smooth curve fitting for the relationship between SFTA1P and IL11.

Table 3 GO terms enriched by 16 DEmRNAs that were involved in the ceRNA network.

Category	Term	P Value	Genes	
GOTERM_BP_FAT	GO:0007167∼ enzyme linked receptor protein signaling pathway	0.0161	CHRDL1, PTPRT, GREM2	
GOTERM_BP_FAT	GO:0030509∼ BMP signaling pathway	0.0257	CHRDL1, GREM2	

RNAs in the ceRNA network are related to survival

LncRNAs, miRNAs and mRNAs associated with prognosis were identified using the expression profiles of 56 lncRNAs, 6 miRNAs and 16 mRNAs in the ceRNA network using Kaplan Meier Survival Curve. After excluding the patients with a follow-up of less than 30 days, ten lncRNAs (PART1, LINC00261, AL163952.1, C2orf48, FAM87A, LINC00052, LINC00472, STEAR-AS1, TSPEAR-AS1 and ERVH48-1) were observed to be significantly related to overall survival rate (P < 0.05) (Fig. 8).

Figure 8 Kaplan–Meiercurve analysis of DElncRNAs and overall survival rate in tongue squamous cellcarcinoma patients.

lncRNAs in the ceRNA network are related to clinical features

The 56 DElncRNAs from the ceRNA network were further analyzed to identify their correlations with clinical features. TSCC patients were divided into subgroups according to pathological stage (Stage III + IV vs. Stage I + II) and TNM stage (T3 + T4 vs. T1 + T2, N2 + N3 vs. N0 + N1). We found six lncRNAs with a high expression level (LINC00355, PSORS1C3, LINC00520, AC112721.1, AL139147.1, SFTA1P) and four lncRNAs with a low expression level (HCG22, LINC00492, AL035696.1, ERVH48-1) were significantly associated with the progression of TSCC (Table 4).

Table 4 The correlations between DElncRNAs in the ceRNA network and clinical characteristics of tongue squamous cell carcinoma.

Comparisons	Downregulated	Upregulated	
Pathologic Stage (Stage III + IV vs. stage I + II)	HCG22, LINC00492, AL035696.1	LINC00355, PSORS1C3	
Pathologic_T (T3 + T4 vs. T1 + T2)	ERVH48-1	LINC00520, PSORS1C3	
Pathologic_N (N2 + N3 vs. N0+ N1)	LINC00492, ERVH48-1	AC112721.1, AL139147.1, SFTA1P	

Discussion

TSCC is the most common form of oral cancer. Dysregulated genes are considered a major cause of oncogenesis and the development of TSCC. Recently, the crucial role of lncRNA in gene expression regulation at three levels including transcription, post-transcription and translation has attracted considerable interest. Accordingly, the ceRNA hypothesis was proposed, postulating that lncRNAs could act as part of post-transcriptional gene expression control. This conclusion generated new insights into the biology of cancer.

To better understand how lncRNA-associated ceRNA crosstalk affects TSCC, we exploited a large-scale TSCC data from the TCGA database and successfully established a dysregulated lncRNA-associated ceRNA network. In addition, growing evidence has indicated that lncRNAs have greater potential as prognostic biomarkers than protein-coding genes due to their stronger correlation with tumor status (Hauptman & Glavac, 2013). Thus, we also identified ten lncRNAs (PART1, LINC00261, AL163952.1, C2orf48, FAM87A, LINC00052, LINC00472, STEAR-AS1, TSPEAR-AS1 and ERVH48-1) as prognostic biomarkers for TSCC. However, there is no research to clearly explain the function of AL163952.1, C2orf48, FAM87A, STEAR-AS1, TSPEAR-AS1 and ERVH48-1.

PART1 is upregulated and its higher expression is associated with poor prognosis in prostate cancer and non-small cell lung cancer (Li et al., 2017b; Sun et al., 2018). Elevated PART1 promotes prostate cancer cell proliferation and inhibits cell apoptosis (Sun et al., 2018). On the contrary, we found that PART1 had lower expression and was negatively correlated to survival rate in TSCC, which was consistent with findings in OSCC (Li et al., 2017c). This may be because PART1 is located on chromosome 5q12, a region that is usually lost in oral squamous cell carcinoma (OSCC) and head and neck squamous cell carcinoma (HNSCC) (Abou-Elhamd & Habib, 2008; Noutomi et al., 2006). In our ceRNA network, low PART1 expression reduced levels of NR3C2 mediated by mir-301b, and reduced expression of NR3C2 promotes tumor cell proliferation, metastasis and epithelial-to-mesenchymal transition (Yang et al., 2018; Yang et al., 2016a; Zhang et al., 2017; Zhao et al., 2018c). In addition, patients with low levels of NR3C2 have a poor prognosis in pancreatic cancer and renal cell carcinoma (Yang et al., 2016a; Zhao et al., 2018c). The relationship between mir-301b and NR3C2 has also been validated in pancreatic cancer (Yang et al., 2016a). Therefore, PART1/mir-301b/NR3C2 axis may be an important mechanism that involves in TSCC development.

LINC00261 is a tumor suppressor positively associated with prognosis in many tumors, such as hepatocellular carcinoma (Zhang, Li & Han, 2018a), endometrial carcinoma (Fang, Sang & Du, 2018) and non-small cell lung (Liu, Xiao & Xu, 2017). Its functions mainly include inhibiting tumor cell proliferation, invasion and metastasis. LINC00261 was also down-regulated in our study. Notably, decreased expression of LINC00261 indicated a better prognosis in TSCC. Our study found LINC00261 regulated the expression of ENPP4 and ENPP5, however, these two genes have not been extensively studied. ENPP2 as their closely-related molecule has been widely reported to participate in tumor development. Low expression of ENPP2 increases reactive oxygen species (ROS) level, and high ROS level could promote tumor cell apoptosis (Cholia et al., 2018; Dawei, Honggang & Qian, 2018). The special function of LINC00261 still needs to be further investigated.

Several existing studies indicate that LINC00472 plays an important role in inhibiting tumor development (Shen et al., 2015; Su et al., 2018; Ye et al., 2018). Our study also showed the similar results that higher expression of LINC00472 was associated with better prognosis. In addition, LINC00472 may regulate the expression of GREM2 by sponging mir-503. GREM2 is an antagonist of bone morphogenetic proteins (BMP) and could activate Notch signaling pathway (Li et al., 2018a) and Wnt/ β-catenin signaling (Wu, Tang & Yuan, 2015), which may be the reasons that GREM2 involves in the development of TSCC. Given that GREM2’s functions have not been studied in cancer, our results need to be verified by experiments.

The function of LINC00052 varies depending on the location of tumors. It acts as a tumor suppressor through inhibiting cell proliferation, invasion and migration in hepatocellular carcinoma (Xiong et al., 2016; Yan et al., 2018; Zhu et al., 2017), whereas it could promote breast cancer growth (Salameh et al., 2017) and gastric cell metastasis and proliferation (Shan et al., 2017). In our study, it may act as an oncogene because it was significantly up-regulated and its high expression indicated a poor prognosis in TSCC. Unfortunately, our ceRNA network failed to find miRNAs that could interact with LINC00052. The function of LINC00052 is needed to be further investigated.

Considering the correlation between DElncRNAs and clinical characteristics, we found 10 DElncRNAs related to pathologic stage, T stage and N stage. Though the function of these lncRNAs are not well investigated, they also may be as therapeutic targets and present a new road to understand the pathogenesis of TSCC. HCG22 was negatively associated with tumor stage in our study. Similarly, Zhao et al. finds a significant inverse correlation between HCG22 and tumor size (Zhao et al., 2018a). Regarding OSCC, HCG22 was found to be correlated with poor survival basing on TCGA database. However, Feng et al. did not find a similar association using their clinical data (Feng et al., 2017). LINC00355 was positively correlated with distant metastasis, lymphatic metastasis and tumor stage, and negatively correlated with prognosis in colon adenocarcinoma (Zhang et al., 2018b). Upregulated LINC00355 was also associated with poor prognosis in prostate cancer (Jiang et al., 2018). In our study, LINC00355 was positively associated with tumor stage, however, it was not identified to be a prognosis biomarker.

Another DElncRNA, SFTA1P has been reported to be a tumor suppressor by inhibiting cell proliferation, invasion and migration in gastric cancer (Ma et al., 2018). It also increases cisplatin chemosensitivity in lung squamous cell carcinoma; similarly, elevated SFTA1P indicates a longer life (Li et al., 2017a). However, we found SFTA1P was up-regulated in TSCC, especially in lymph node metastasis tumor. Thus, we speculate that SFTA1P may act as an oncogene. Our ceRNA network indicated that SFTA1P up-regulated the expressions of IL-11 or HOXC8 by binding mir-211, and elevated IL-11 or HOXC8 contributes to development of cancer , such as breast cancer (Cai et al., 2018; Li et al., 2014) and non-small cell lung cancer (Liu et al., 2018; Zhao et al., 2018b). Therefore, it is reasonable that our results are not consistent with other studies. Though the functions of IL-11 and HOXC8 are not validated in TSCC, our results also offer new ideas for the development of TSCC.

Our study identified some valuable lncRNAs that are associated with carcinogenesis and survival. Few of them have been validated in vitro and in vivo; however, none of the lncRNAs were validated in TSCC. Hence, these valuable lncRNAs still need to be verified, and our ceRNA network, which was build based on high-throughput sequencing, requires further verification.

Conclusion

Taken together, we identified aberrantly expressed mRNAs, lncRNAs, and miRNAs and then successfully constructed a functional ceRNA network for TSCC tumorigenesis. Key lncRNAs should be check closely for association with survival and clinical features in TSCC patients, which provide novel lncRNAs as potential prognosis biomarkers and therapeutic targets.

Supplemental Information

Table S1 The list of DEmRNAs

Click here for additional data file.

Table S2 Gene ontology analyses of the DEmRNAs according to their biologicalprocess, cellular component and molecular function.

Click here for additional data file.

Table S3 The list of DElncRNAs

Click here for additional data file.

Table S4 The list of DEmiRNAs

Click here for additional data file.

Table S5 Normalized expression matrix of mRNA

This table recorded the normalized expression signal of mRNA of all included samples. The data were normalized.

Click here for additional data file.

Table S6 Normalized expression matrix of lncRNA

This table recorded the normalized expression signal of lncRNA of all included samples. The data were normalized.

Click here for additional data file.

Table S7 Normalized expression matrix of miRNA

This table recorded the normalized expression signal of miRNA of all included samples. The data were normalized.

Click here for additional data file.

Supplemental Information 1 Heatmaps of the DEmRNAs, DElncRNAs and DEmiRNAs

Click here for additional data file.

Figure S4 Venn diagram of mRNAs involved in ceRNA regulation network

The red area presents only the DEmRNAs number instead of the target number. The blue area presents only the target number, rather than the DEmRNAs number, while the purple area in the middle indicates the number of mRNA which is both the differential expression and the target.

Click here for additional data file.

We thank The Cancer Genome Atlas (TCGA) project and its contributors for this valuable public data set.

Additional Information and Declarations

Competing Interests

Author Contributions

Data Availability

The authors declare there are no competing interests.

Shusen Zhang conceived and designed the experiments, performed the experiments, analyzed the data, contributed reagents/materials/analysis tools, authored or reviewed drafts of the paper.

Ruoyan Cao conceived and designed the experiments, performed the experiments, analyzed the data, contributed reagents/materials/analysis tools, prepared figures and/or tables, authored or reviewed drafts of the paper.

Qiulan Li performed the experiments, analyzed the data.

Mianfeng Yao analyzed the data.

Yu Chen prepared figures and/or tables.

Hongbo Zhou authored or reviewed drafts of the paper, approved the final draft.

The following information was supplied regarding data availability:

Raw data is available in the Supplemental Materials.

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
