# Peer review of "Comprehensive analysis of lncRNA-associated competing endogenous RNA network in tongue squamous cell carcinoma"

_PeerJ, doi:10.7717/peerj.6397_

## Round 0.1 · original submission · Major Revisions

Both the reviewers have raised critical questions related to this study. Although this study is highly relevant for carcinogenesis, both major and minor concerned has to be addressed to improve the quality of this article.

·

Basic reporting

The manuscript entitled “Comprehensive analysis of lncRNA-associated competing endogenous RNA network in tongue squamous cell carcinoma” is a well-written manuscript. Language polishing will be much appreciated. In the current study, the authors used existing sequencing data to investigate differentially expressed lncRNA, mRNA and miRNA and used this information to construct competitive endogenous RNA (ceRNA) network. This study will defiantly help in understanding the RNA interaction network and mechanism in TSCC.

Experimental design

no comment

Validity of the findings

no comment

Additional comments

Major

1) Flowchart of the ceRNA network construction must be included for better understanding of the approach used
2) Gene set enrichment analysis may add value to this article
3) In the results section, line 183, 184, 185 need to be clear and explained

Minor

1) The conclusion in the abstract can be clearer and specific
2) Supplementary figure 4 legend should be clearer and written better
3) Must include expansion for MRE in line 73
4) Paragraph transitions need to be improved in the introduction
5) Resolution of figure need to improve especially for fig 7
6) Space needs to add in between (TSCC) and remains limited

Reviewer 2 ·

Basic reporting

The authors identified lncRNAs as potential prognostic and therapeutic targets for TSCCs. They analyzed RNA expression profiles from tumor and normal control tissues and identified dysregulated mRNAs, lncRNAs and microRNAs. Importantly, they built a competing endogenous RNA (ceRNA) network that would be useful tool in predicting disease survival. At few places the English syntax needs to be checked. The literature review is not sufficient. Some figures have very small text making them difficult to read.

Experimental design

In this study, Zhang et al utilize the publicly available dataset from the TCGA portal, after filtering criteria chose 122 patients and 15 normal controls for the study. They identified the differentially expressed mRNAs, lncRNAs and miRNAs in TSCCs and generated ceRNA networks that were related to survival.

Validity of the findings

Although highly relevant in studying carcinogenesis, this study lacks a strong hypothesis, rigorous interpretation, and exhaustive analysis, and the inference is restricted to just one type of cancer. This manuscript needs to incorporate broader application of the ceRNA networks they identified, use additional statistical analyses and expand the methodology to link the identified lncRNAs to carcinogenesis and survival.

Additional comments

The authors identified lncRNAs as potential prognostic and therapeutic targets for TSCCs. They analyzed RNA expression profiles from tumor and normal control tissues and identified dysregulated mRNAs, lncRNAs and microRNAs. Importantly, they built a competing endogenous RNA (ceRNA) network that would be useful tool in predicting disease survival.
In this study, Zhang et al utilize the publicly available dataset from the TCGA portal, after filtering criteria chose 122 patients and 15 normal controls for the study. They identified the differentially expressed mRNAs, lncRNAs and miRNAs in TSCCs and generated ceRNA networks that were related to survival.
Although highly relevant in studying carcinogenesis, this study lacks a strong hypothesis, rigorous interpretation, and exhaustive analysis, and the inference is restricted to just one type of cancer. This manuscript needs to incorporate broader application of the ceRNA networks they identified, use additional statistical analyses and expand the methodology to link the identified lncRNAs to carcinogenesis and survival. Additionally, the authors need to address following minor concerns:
• #34 lncRNA not lnRNA
• #72 Delete In 2011
• #148 heat map of?
• Figure 2A text is not legible
• Authors need to explain the relevance of their study as only two lncRNAs showed significant relation to overall survival
• Authors should investigate and report if the lncRNAs in the ceRNA network they identified show any correlation with clinical features of other types of cancers
• A schematic explaining the workflow could be useful

---

## Round 0.2 · accepted · Accept

Authors have addressed all the questions raised by both the reviewers. Manuscript can be accepted for publication.

# ·

Basic reporting

I am satisfied with the revision. All the text in the figures should be of the same font. Maintain it with same text size where ever possible. Please check carefully and modify

Experimental design

No comment

Validity of the findings

No comment

Additional comments

All the text in the figures should be of the same font. Please check carefully and modify

I am satisfied with the revision.

Reviewer 2 ·

Basic reporting

The authors have adequately acknowledged and addressed the suggestions on the experiments and data representation previously raised by the reviewer. The grammar and syntax has also been significantly improved. The manuscript can be published.

Experimental design

Experimental design has been promptly explained and represented.

Validity of the findings

The competing endogenous RNA (ceRNA) network that the authors have built would be useful tool in predicting disease survival.

Additional comments

The authors have adequately acknowledged and addressed the suggestions on the experiments and data representation previously raised by the reviewer. The grammar and syntax has also been significantly improved. The manuscript can be published.